# Classification of Radar Targets with Micro-Motion Based on RCS Sequences Encoding and Convolutional Neural Network

Xuguang Xu *, Cunqian Feng and Lixun Han

Air and Missile Defense College, Air Force Engineering University, Xi'an 710038, China
* Correspondence: xuxuguanghjbsqsbs@163.com

**Abstract:** Radar cross section (RCS) sequences, an easy-to-obtain target feature with small data volume, play a significant role in radar target classification. However, radar target classification based on RCS sequences has the shortcomings of limited information and low recognition accuracy. In order to overcome the shortcomings of RCS-based methods, this paper proposes a spatial micro-motion target classification method based on RCS sequences encoding and convolutional neural network (CNN). First, we establish the micro-motion models of spatial targets, including precession, swing and rolling. Second, we introduce three approaches for encoding RCS sequences as images. These three types of images are Gramian angular field (GAF), Markov transition field (MTF) and recurrence plot (RP). Third, a multi-scale CNN is developed to classify those RCS feature maps. Finally, the experimental results demonstrate that RP is best at reflecting the characteristics of the target among those three encoding methods. Moreover, the proposed network outperforms other existing networks with the highest classification accuracy.

**Keywords:** micro-motion; RCS sequences encoding; target classification; multi-scale CNN

## 1. Introduction

Micro-motion is an important feature of space targets, which is of great significance for target parameters estimation [1,2]. Over the past decade, scholars have carried out extensive research on the micro-motion of space radar targets, and relevant results have been successfully applied to the characteristic analysis of satellites, ballistic targets (BTs), space debris and other targets [3–5].

BT classification is a typical application of micro-motion [6–9]. The micro-motion of the warhead is precession due to the release of the decoy. Lacking an attitude control system, the micro-motion of the heavy decoy is swing, and the micro-motion of the light decoy is rolling. Furthermore, the micro-motions of boosters and the debris are usually rolling. The micro-motions of BTs are different and unique [7,10,11], providing an effective tool for BT classification.

Scholars have conducted various research works on the radar echoes received from BTs. The micro-Doppler (MD) time–frequency graphs (TFGs), the high-resolution range profile (HRRP) and the high-resolution range profile sequences (HRRPs) and the RCS sequences are three important features of echoes and have been widely used for discriminating BTs [12]. Correspondingly, BT classification methods based on micro-motion features are generally classified into the following three categories. The first class mainly uses the MD features on the TFGs of the target. Choi extracted the fundamental frequency, the bandwidth and the sinusoidal moment of TFGs to distinguish the precession/nutation warheads from the wobble decoys [13]. Jung designed a CNN to classify warheads and decoys based on TFGs and a cadence velocity diagram [14]. Zhang presented a complex-valued coordinate attention network to classify the cone–cylinder precession targets with different MD parameters [15]. The second category mainly depends on HRRP/HRRPs to discriminate BTs. Zhou employed HRRPs to estimate the parameters of BTs and thus

realized the classification of the warhead, the heavy decoy and the light decoy according to the differences of their parameters [16]. Persico conducted an Iradon transform on HRRP to generate a new feature and input the feature into the K-nearest neighbor classifier (KNN) to classify warheads and false targets [17]. Wang created a novel processing flow based on HRRPs and used CNN to classify five types of micro-motions with similar shapes [18]. The third category mainly relies on RCS sequences. Cai discriminated warheads from debris based on several statistics of RCS sequences, such as the expectation, the variance and the period [19]. Ye developed a gated recurrent unit (GRU) network for radar target recognition based on several statistics of RCS sequences [20]. Choi proposed a multi-feature fusion framework based on RCS sequence for BTs recognition [12]. Chen input RCS sequences into a one-dimensional CNN to realize the intelligent identification of warheads and decoys [21]. In addition to the above three types, methods based on feature fusion have also been a promising technique [22]. Chen fused TFGs and HRRPs of the flying bird and drone to generate a new feature map and then input them to a modified multi-scale CNN to classify the targets [4]. Features based on a combination of RCS, TF, HRRP and RID were adopted to achieve a decision-level fusion, and thereby, a high-accuracy recognition was achieved for space targets with micro-motion [23].

Although the above methods can detect warheads from BTs, there are some significant drawbacks that limit their widespread application. (1) HRRP/HRRPS can only be acquired by wideband radars, and narrowband radars do not have this capability. (2) Only when the repetition frequency of the radar is greater than twice that of the MD can the valid TFGs be generated. Otherwise, Doppler ambiguity will occur in TFGs. (3) The information reflected by RCS sequences is relatively abstract, resulting in a low accuracy for the classification task. Therefore, research works on simple and high-precision methods for BTs recognition have drawn ever-increasing attention.

RCS sequences are easily processed due to the small data volume. Moreover, both wideband radars and narrowband radars can acquire RCS sequences. Therefore, how to mine advanced features from RCS sequences has become the key point to promote the application of RCS-based methods. This paper proposes a BTs classification structure based on RCS sequence encoding and multi-scale convolutional neural network. We introduce MTF, GAF and RP to convert RCS sequences to images, so as to improve the richness of RCS sequences. We develop a multi-scale CNN to extract the features of these images and discriminate four different BTs. Moreover, several experiments are conducted to evaluate the performance of the encoding methods and the proposed network.

The rest of this paper is organized as follows. Section 2 presents the micro-motion of different BTs. Section 3 describes three encoding methods for RCS sequences. A multi-scale CNN is proposed in Section 4. Section 5 comprises the simulation results and performance analysis. Section 6 is the conclusion.

## 2. The Micro-Motion Model of BTs

Before establishing the micro-motion model, we first introduce the concept of RCS. RCS is a measure of an object's ability to reflect the electromagnetic wave, defined as $4\pi$ times that of the ratio of the reflected power of the target to the incident power. Set RCS as $\sigma$, and the definition is written as

$$\sigma = \lim_{R \to \infty} 4\pi R^2 \left| \frac{E_s}{E_i} \right|^2 = 4\pi \frac{\left| E_{far,\,H} \right|^2 + \left| E_{far,\,V} \right|^2}{|E_i|^2} = \sigma_H + \sigma_V \qquad (1)$$

where $E_i$ represents the intensity of the incident electric field from the radar, $E_s$ represents the intensity of the scattering electric field from the target, $R$ is the range from the radar to the target, $R \to \infty$ indicates that the observation scene is a far-field condition, and the incident wave is regarded as the uniform plane wave. $E_{far,\,H}$ and $E_{far,\,V}$ represent the electric field horizontal component and vertical component, respectively.

Micro-motion will change the attitude of the target, causing fluctuations in RCS. According to the change of RCS sequences, we can derive the micro-motion type and the shape of the target [20]. The above description is the basis of the recognition methods based on RCS sequences.

Rotationally symmetric structures are commonly adopted for BTs, including cone, cone–cylinder, ellipse and so on. For these shapes, the angle between the symmetry axis of the target and the radar line of sight (LOS) (defined as the aspect angle) determines the value of RCS.

In the remainder of this section, we first establish the models of different micro-motions in Figure 1 and then derive the expression of the aspect angle for every motion.

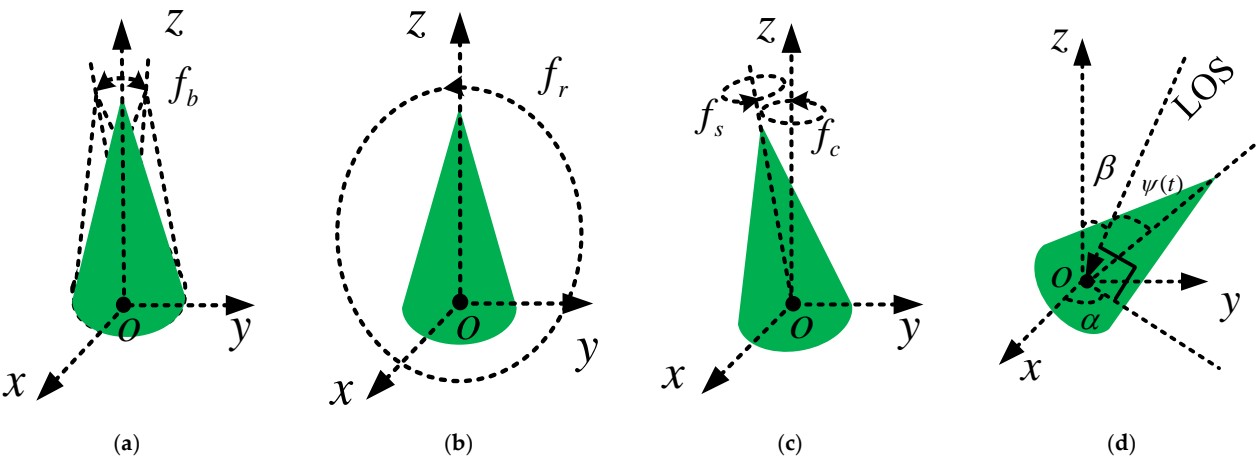

**Figure 1.** (**a**) Swing; (**b**) Rolling; (**c**) Precession; (**d**) Observation models of different micro-motions.

Swing: As shown in Figure 1a, the target swings at a small angle along $oz$ in $yoz$. Set the amplitude of swing as $\theta_b$, the initial phase as $\theta_{b\_0}$ and the swing frequency as $f_b$. Setting the angle between the symmetry axis and $oz$ as $ang_{sw}(t)$, we denote $ang_{sw}(t)$ as

$$ang_{sw}(t) = \theta_b \sin(2\pi f_b t + \theta_{b\_0}) \tag{2}$$

Based on $ang_{sw}(t)$, the unit vector of the symmetry axis is given as

$$n_{sw}(t) = [0,\ \sin(ang_{sw}(t)),\ \cos(ang_{sw}(t))] \tag{3}$$

Rolling: As shown in Figure 1b, the target rolls in $yoz$. Set the initial phase as $\theta_{r\_0}$ and the rolling frequency as $f_r$. Set the angle between the symmetry axis and $oz$ as $ang_{rol}(t)$, and it is written as

$$ang_{rol}(t) = \theta_{r\_0} + 2\pi f_r t \tag{4}$$

According to Equation (3), the unit vector of the symmetry axis is written as

$$n_{rol}(t) = [0,\ \sin(ang_{rol}(t)),\ \cos(ang_{rol}(t))] \tag{5}$$

Precession: As shown in Figure 1c, precession is a combination of spinning and conning. For rotationally symmetric targets, spinning does not modulate the echo. Therefore, only the modulation of the conning is considered. Set the precession frequency as $f_p$ and the initial phase as $\theta_{p\_0}$. The azimuth angle of the symmetry axis in $yoz$ is written as

$$ang_{pre}(t) = 2\pi f_p t + \theta_{p\_0} \tag{6}$$

Set the precession angle as $\theta_p$, and the unit vector of the symmetry axis is expressed as

$$n_{pre}(t) = [\cos(ang_{pre}(t))\sin(\theta_p),\ \sin(ang_{pre}(t))\sin(\theta_p),\ \cos(\theta_p)] \tag{7}$$

Moreover, we also establish the observation model in Figure 1d. Set $\alpha$ as the azimuth angle of the LOS and the angle between LOS and $oz$ as $\beta$. The unit vector of the LOS is expressed as

$$n_{LOS} = [\cos\alpha\sin\beta, \ \sin\alpha\sin\beta, \ \cos\beta] \tag{8}$$

Set $\psi(t)$ as the average aspect angle, and it is solved as

$$\psi(t) = cos^{-1}(n_{LOS} \times n_s(t)) \tag{9}$$

where

$$n_s(t) \in [n_{sw}(t), \ n_{rol}(t), \ n_{pre}(t)]$$

## 3. Methods for RCS Sequences Encoding

The raw RCS sequence is a 1D signal, which has a relatively poor ability to present the targets' characteristics and is not conducive to target classification. We believe that converting RCS sequences to 2D images can show the features more abundantly and intuitively, which is an effective and efficient approach to visualize the details and features. In the radar signal processing field, the time–frequency analysis is the most popular encoding method. However, time–frequency analysis has high requirements for sequence length and repetition frequency, so its application scope is very limited.

Based on the above analysis, we introduce three time series encoding methods to map RCS into two-dimensional space, so as to enhance the feature representation of RCS and lay the foundation for high-precision target recognition. These three methods are MTF, GAF and RP. As far as we know, the research on RCS analysis using these three time series coding methods is very limited, and this is exactly one of the important contributions of this paper.

### 3.1. MTF

The MTF can effectively capture the state transition information of the time series, thereby improving the feature expression [24,25].

Denoting $Y = (y_1, y_2, \cdots, y_n, \cdots, y_N), n \in [1, \ N]$ as a RCS sequence with the length being $N$, identify $Q$ quantile bins of $Y$, and every $y_n$ is assigned into the related bins $q_i(i \in [1, \ Q])$. Counting the first-order transitions among the different bins along the time axis, we will obtain a $Q \times Q$ weighted adjacency matrix $W$, where $w_{i,j}$ is the normalized transition probability of $q_i \rightarrow q_j$. Nevertheless, $W$ is insensitive to the step size of the time series. To enhance the expressiveness of temporal information, MTF is designed based on $W$, and the expression is shown as

$$M = \begin{bmatrix} w_{i,j}\big|_{y_1\in q_i,y_1\in q_j} & \cdots & w_{i,j}\big|_{y_1\in q_i,y_N\in q_j} \\ \vdots & \ddots & \vdots \\ w_{i,j}\big|_{y_N\in q_i,y_1\in q_j} & \cdots & w_{i,j}\big|_{y_N\in q_i,y_N\in q_j} \end{bmatrix} \tag{10}$$

Compared to $W$, $M$ contains not only the step size information of the time series but also the state transition information. As a result, MTF is widely used for time series analysis.

### 3.2. GAF

GAF displays the temporal correlation of the sequence in a 2D image, where the motion information of the sequence is represented as the change from the upper left corner to the lower right corner [26,27]. GAF is based on a transformation from the Cartesian coordinate system to the polar coordinate system. The generation process mainly includes three steps.

Step 1. Rescale $Y$ to $\widetilde{Y}$ via (11), so that $y_n$ falls between $-1$ and $1$.

$$\widetilde{y}_n = \frac{(y_n - \max(Y)) + (y_n - \max(Y))}{\max(Y) - \min(Y)} \tag{11}$$

Step 2. Transform the scaled sequence from the Cartesian coordinate system $(t, y_n)$ to the polar coordinate system $(r_n, \phi_n)$ via Equation (12).

$$\begin{cases} \phi_n = \arccos(\widetilde{y}_n) \\ r_n = \frac{t_n}{N_0} \end{cases} \tag{12}$$

where $t_n$ denotes the time stamp, and $N_0$ is a constant factor to regularize the span of the polar coordinate system.

Step 3. Extract the angle information in polar coordinates and generate GAF via Equations (13) and (14).

$$GASF = \begin{bmatrix} \cos(\phi_1 + \phi_1) & \cdots & \cos(\phi_1 + \phi_N) \\ \vdots & \ddots & \vdots \\ \cos(\phi_N + \phi_1) & \cdots & \cos(\phi_N + \phi_N) \end{bmatrix} \\ = \widetilde{Y}^{\mathrm{T}}\widetilde{Y} - \left(\sqrt{I - \widetilde{Y}^2}\right)^{\mathrm{T}} \sqrt{I - \widetilde{Y}^2} \tag{13}$$

$$GADF = \begin{bmatrix} \sin(\phi_1 - \phi_1) & \cdots & \sin(\phi_1 - \phi_N) \\ \vdots & \ddots & \vdots \\ \sin(\phi_N - \phi_1) & \cdots & \sin(\phi_N - \phi_N) \end{bmatrix} \\ = \left(\sqrt{I - \widetilde{Y}^2}\right)\widetilde{Y} - \widetilde{Y}\left(\sqrt{I - \widetilde{Y}^2}\right) \tag{14}$$

where GASF denotes the Gramian angular summation fields, GADF denotes the Gramian angular difference fields, $I$ represents the identity matrix, $(\cdot)^{\mathrm{T}}$ represents the transpose of the elements.

GASF presents the correlation between two moments based on the sum of cosine functions, while GADF presents the correlation by the difference of sine functions. After conducting GAF on the time series, the correlations of the time series are significantly enhanced.

### 3.3. RP

As a nonlinear system analysis tool, RP could detect nonlinear features and visualize the recurrent behavior in the time series [28,29].

Set the delay time as $\tau$, the embedding dimension as $m$, and the phase space matrix of the time series is calculated as

$$Z = [z_1, z_2, \cdots, z_M] = \begin{bmatrix} y_1 & y_2 & \cdots & y_M \\ y_{1+\tau} & y_{2+\tau} & \cdots & y_{M+\tau} \\ \vdots & \vdots & & \vdots \\ y_{1+(m-1)\tau} & y_{2+(m-1)\tau} & \cdots & y_{M+(m-1)\tau} \end{bmatrix} \tag{15}$$

where

$$M = N - (m-1)\tau$$

Denoting $R$ as the RP, $R$ can be solved as

$$R(p, q) = \Theta(\varepsilon - \|z_p - z_q\|) = \begin{cases} 1; \varepsilon \geq \|z_p - z_q\| \\ 0; \varepsilon < \|z_p - z_q\| \end{cases} \tag{16}$$

where $\|\cdot\|$ denotes a norm, and the $L_2$ norm is selected in this paper; $\varepsilon$ is a threshold to determine the state of RP, and $p, q \in [1, M]$.

The generation of RP involves three parameters: $\tau$, $m$ and $\varepsilon$. As for the selection of these three parameters, this paper follows the following rules.

(1) $\varepsilon$. There is a disagreement over the selection of $\varepsilon$. According to some scholars, $\varepsilon$ is related to the signal-to-noise ratio (SNR), and according to others, some $\varepsilon$ is related to the phase space radius. An unsuitable $\varepsilon$ can easily lose the details of the sequence. To preserve

the details of the RP, this paper improves the RP in Equation (16) into a non-threshold RP without taking $\varepsilon$ into consideration. The non-threshold RP is expressed as

$$R(p,q) = \left\| y_p - y_q \right\|_2^2 \tag{17}$$

(2) $\tau$ and $m$. Many methods have been utilized to select these two parameters, but different methods may yield different results [30]. After consulting many literature works, we have still not found an acceptable method. Therefore, different values of $\tau$ and $m$ are empirically selected to test the performance based on Refs. [28,31–33].

These three encoding methods have their own characteristics, and each method contains different information of the RCS sequence. The MTF emphasizes the state transition of the sequences; GAF is good at representing the temporal relation; and RP is able to display the chaos characteristics. By evaluating the classification performance of these methods, the most suitable encoding method can be determined.

## 4. Proposed Network

With the development of the deep-learning theory, CNNs have been widely used in image classification, pattern recognition and parameter regression [34]. Instead of manually extracting the features, CNNs learn the advanced features by themselves and show advanced performance in those tasks. Typical CNNs include Alexnet, Googlenet and Resnet [4,35]. To achieve an efficient classification, this paper designs a multi CNN. The details of the network are as follows.

### 4.1. Res2Net

ResNet is a commonly used module in CNN to solve the gradient problem (shown in Figure 2a). On the basis of ResNet, Gao developed a multi-scale module called Res2Net by building hierarchical residual-like connections within one single residual block [36]. Res2net is an effective multi-scale technique, which further explores the multi-scale features and extends the range of receptive fields [37].

The structure of Res2Net is shown in Figure 2b. For the output feature map of the previous layer, the $1 \times 1$ convolution is adopted to adjust the number of channels; then, the feature map is split into $s$ feature map subsets $x_i, i \in [1\ s]$, and $s$ is the scale factor of Res2Net. The process of channel splitting and convolution can be formulated as

$$y_i = \begin{cases} x_i & i = 1 \\ K_i(x_i) & i = 2 \\ K_i(x_i + y_{i-1}) & 2 < i \le s \end{cases} \tag{18}$$

where $K_i(\cdot)$ represents the $3 \times 3$ convolution for $x_i$. In addition to $x_1$, each $x_i$ is summed with the output of the previous layer, and the obtained feature map is input into a $3 \times 3$ convolution to obtain $y_i$. $K_i(\cdot)$ could receive the feature from other scales before $i$th, making Res2Net possess a greater receptive field. Such structure can effectively capture the global and local features of the feature maps, so that the feature extraction ability is improved.

Finally, we stack $y_i$ along the channel dimension and input the stacked feature map into a $1 \times 1$ convolution to match the size of the input.

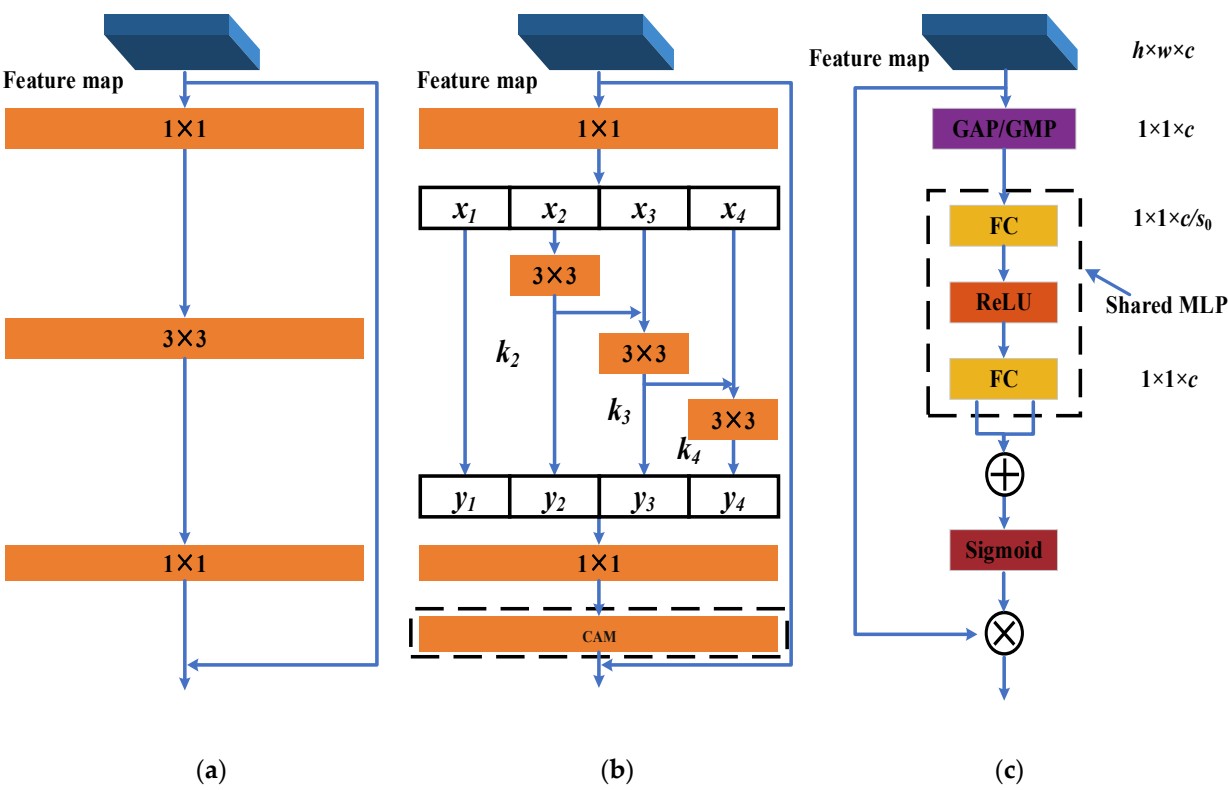

**Figure 2.** The Res2net module and CAM module. (**a**) Resnet module; (**b**) Res2net/CAM–Res2net module; (**c**) CAM module.

### 4.2. Channel Attention

To take advantage of these features from different channels, Woo developed a module called the channel attention module (CAM) [38]. CAM assigns attention weights to every channel of the input to obtain a more effective feature map. The structure of CAM is shown in Figure 2c.

For the input $(W \times H \times C)$, a global average/maximum pooling layer is first conducted along the channel dimension to obtain a vector $(1 \times 1 \times C)$. Then, the vector is transformed into the attention weights by a shared MLP (containing two fully connected (FC) layers and a ReLU activation function) and a Sigmoid function. The first FC is utilized to compress the vector $(1 \times 1 \times C/s)$, and the second FC is used to restore the vector $(1 \times 1 \times C)$. $s$ is the squeeze ratio and is set to 16 in this paper.

CAM can be plugged into the Res2net to fuse the channel-wise information, and thereby, the CAM–Res2net module is generated.

### 4.3. The Activation Function (AF) and the Loss Function

The most regular AF in CNNs is the ReLU AF, which is very simple to solve. For input $x_f$, the output through the ReLU AF is given as

$$\text{ReLU}(x_f) = \max(0, \ x_f) \tag{19}$$

The ReLU AF will output the negative samples of $x_f$ as 0, thereby destroying the transmission of information. The Mish AF is a novel AF used in YOLOv4 for object detection [39]. Based on a combination of the Tanh AF and the Softplus AF, the Mish AF is expressed as

$$\text{Mish}(x_f) = x_f \times \tanh(\ln(1 + e^{x_f})) \tag{20}$$

The Mish AF is a continuous function, which has a nonlinear representation effect on positive values and does not induce gradient disappearance on negative values. The Mish

AF has a strong regularization limitation and can effectively improve the feature expression ability of the model. However, the complexity of the Mish AF will consume the computing power of the computer. Therefore, both the Mish AF and the ReLU AF are contained in the proposed network, where the Mish AF is used for the deep features, and the ReLU AF is used for the shallow features.

Moreover, we choose the cross entropy loss as the cost function of the proposed network. The cross entropy loss is given as

$$L = -\sum_n p_n \log(c_n) + (1 - p_n) \log(1 - c_n) \tag{21}$$

where $p_n$ and $c_n$ represent the predicted label and the real label, respectively.

In summary, the structure of the proposed network is shown in Table 1.

**Table 1.** The configuration of the proposed multi-scale CNN for BTs classification.

| Order | Shortcut | Cycle | Operation | Output Size |
|---|---|---|---|---|
| 1 | — | — | Input | $64 \times 64 \times 1$ |
| 2 | — | — | Conv 32.3 $\times$ 3, stride = 1 | $62 \times 62 \times 32$ |
| 3 | — | — | Conv 128.3 $\times$ 3, stride = 1 | $62 \times 62 \times 128$ |
| 4 | — | — | Maxpooling, 3 $\times$ 3, stride = 2 | $30 \times 30 \times 128$ |
| 5 | — | — | Res2net 128-256-128 | $30 \times 30 \times 128$ |
| 6 | Conv 1 $\times$ 1, stride = 2 | — | DS-Conv 768.3 $\times$ 3, stride = 1 DS-Conv 768.3 $\times$ 3, stride = 1 Maxpooling, 3 $\times$ 3, stride = 2 | $15 \times 15 \times 768$ |
| 7 | — | 2 | Res2net | $15 \times 15 \times 768$ |
| 8 | — | — | CAM-Res2net | $15 \times 15 \times 768$ |
| 9 | Conv 1 $\times$ 1, stride = 2 | — | DS-Conv 1024.3 $\times$ 3, stride = 1 DS-Conv 1024.3 $\times$ 3, stride = 1 Maxpooling, 3 $\times$ 3, stride = 2 | $8 \times 8 \times 1024$ |
| 10 | — | — | CAM-Res2net | $8 \times 8 \times 1024$ |
| 11 | — | — | DS-Conv 256.3 $\times$ 3, stride = 1 | $8 \times 8 \times 256$ |
| 12 | — | — | Global Avgpooling | $1 \times 1 \times 256$ |
| 13 | — | — | Dropout 0.2 | $1 \times 1 \times 256$ |
| 14 | — | — | Fc 4 | $1 \times 1 \times 4$ |
| 15 | — | — | Output | Predicted label |

It needs to be noted that regular convolution is replaced with depthwise separable convolution (DS-Conv) due to fewer parameters. Because DS-Conv is an existing technique, the details are presented in the Appendix A instead of the main text.

As for the motivation of the configuration in Table 1, we will explain as follows.

Several research works have proven that the multi-scale CNN performs well in the classification task. As a result, we propose a multi-scale CNN to achieve high-accuracy classification in this manuscript. Res2net, channel attention, DS-Conv and other details form the proposed network. Actually, the configurations of the existing classical CNNs provide us with a meaningful reference, and those configurations are selected based on the existing CNNs and our own understanding of the CNN's configuration. Some details are explained as follows.

Due to fewer parameters and more nonlinearity, the kernel size of the convolution is set to 3 $\times$ 3 instead of 5 $\times$ 5, 7 $\times$ 7 and other sizes. There are more convolution kernels in the deeper position of the network (changing from 32, 64, 768, 1024), which shows the emphasis on deeper features. Due to the better validity and reliability of the deeper features, several Res2net/CAM–Res2net modules are adopted to learn the deeper features instead of the shallow features. Ds-Conv is adopted to reduce the number of parameters. Maxpooling and avgpooling are adopted to reduce the dimension of the feature map and strengthen the nonlinearity of the network.

In fact, the convolution layer is always followed by a batch normalization layer and a nonlinear activation layer in this paper. It should be noted that the batch normalization layer and nonlinear activation layer are not exhibited in Table 1. Moreover, the ReLU function exists in parts 1–7, and the Mish function exists in the rest of the network.

## 5. Experiments

According to Refs [20,21,40], we establish the following four geometric models to represent the warhead, the heavy decoy, the light decoy and the booster (Figure 3).

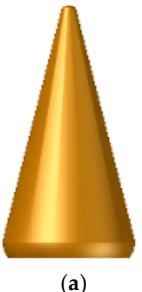
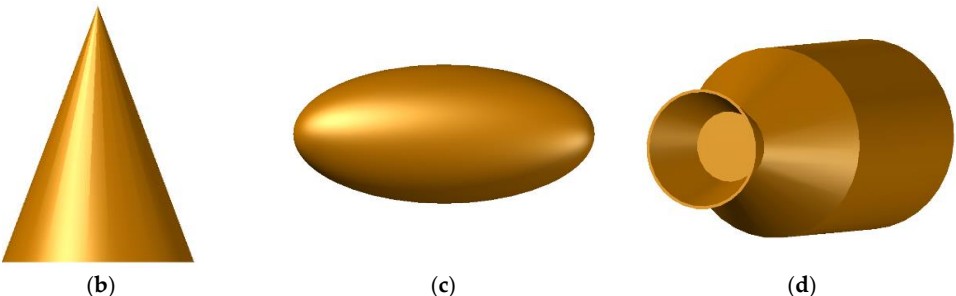

(a)          (b)          (c)          (d)

**Figure 3.** Models of the BTs. (**a**) Warhead; (**b**) Heavy decoy; (**c**) Light decoy; (**d**) Booster.

In the classification task for micro-motion targets, it is difficult to classify objects with similar motions and similar structures. In this paper, the warhead's structure is similar to that of the weight decoy's, and the light decoy's motion type is similar to that of the booster's. This configuration is used to verify that the proposed network can effectively address this classification difficulty.

The micro-motion parameters of the above targets are shown in Table 2.

**Table 2.** The parameter settings of the datasets.

|  | Warhead Precession | Heavy Decoy Swing | Light Decoy Rolling | Booster Rolling |
|---|---|---|---|---|
| $\alpha(\circ)$ | 20 : 20 : 60 | 20 : 20 : 60 | 15 : 10 : 65 | 15 : 10 : 65 |
| $\beta(\circ)$ | 30 : 10 : 60 | 20 : 15 : 155 | 20 : 15 : 155 | 15 : 10 : 65 |
| $f_c(\text{Hz})$ | 0.5 : 1 : 3.5 | − | − | − |
| $f_b(\text{Hz})$ | − | 0.5 : 1 : 3.5 | − | − |
| $f_r(\text{Hz})$ | − | − | 0.5 : 0.4 : 3.3 | 0.5 : 0.4 : 3.3 |
| $\theta_p(\circ)$ | 6 : 1 : 15 | − | − | − |
| $\theta_b(\circ)$ | − | 6 : 3 : 15 | − | − |

For each type of target in Table 2, there are 480 groups of different parameters to generate the RCS of the target.

For BTs classification, datasets based on the measured data are difficult to obtain. As a result, datasets based on electromagnetic calculation are always adopted in these research works [15]. This paper also uses electromagnetic calculation to simulate RCS sequences. The generation process of the training dataset is shown as follows.

Step 1. Based on the real structure of BTs, we establish the simplified geometric model by using AutoCAD (Autodesk Computer Aided Design, a drawing tool software for 2D drawing, detailed drawing and basic 3D design).

Step 2. We import the geometric model into FEKO [20,41] (a software widely used for electromagnetic calculation) to solve the RCS. Set the source as plane wave source, the observation condition as far field, the frequency as 8GHz (this setting can be thought of as the signal type of the radar being a single frequency signal) and the polarization as the vertical linear polarization. Moreover, the physical optics (PO) algorithm is selected as the electromagnetic calculation method in FEKO.

Step 3. Based on the configuration above, we change the value of the pitching angle of the incident wave from $0°$ to $180°$ with a step size of $0.001°$ and solve the static RCS $\sigma_{0-180}$ of the object.

It needs to be noted that because the polarization of the incident wave is vertical linear polarization, we can thus rewrite Equation (1) as the following equation to solve $\sigma_{0-180}$.

$$\sigma_{Total} \approx \sigma_{Vertical} = 4\pi \frac{\left|E_{far,\,V}\right|^2}{\left|E_0\right|^2} \tag{22}$$

Step 4. According to the micro-motion parameters in Table 2, we solve the $\psi(t)_{t=0-3s}$ via Equations (2)–(9) for each group parameter of the target.

Moreover, during the calculation of $\psi(t)_{t=0-3s}$, the repetition frequency of the radar is set to 256 Hz, and the duration time is 3 s.

Step 5. Based on the value of $\psi(t)_{t=0-3s}$ and $\sigma_{0-180}$, the method of interpolation is employed to solve the RCS sequence $\sigma_{\psi(t)}$.

Step 6. For every $\sigma_{\psi(t)}$, two sequences will be extracted with stage 0–2 s and stage 0.25–2.25 s. The length of one RCS sequence is 512.

Step 7. Perform different encoding methods on the RCS sequence, and hence, the encoded images are generated. Adjust the size of the encoded image to $64 \times 64 \times 1$ to match the input size of the proposed network, and thereby, the dataset for BT classification is generated.

As a result, there are four types of micro-motion targets in the dataset, and the number of samples for each type is 960. If we want to add noise to the RCS, the noise should be added to the $E_{far,\,V}$ and then solve the value of $\sigma_{\psi(t)}$ via Equation (22). For each fixed SNR, there are 3840 samples in the dataset.

To visualize the modulation of micro-motion on RCS sequences and on encoded images, we randomly selected one sample for every target. The RCS sequences and the corresponding encoded images of the four targets are shown in Figures 4–7.

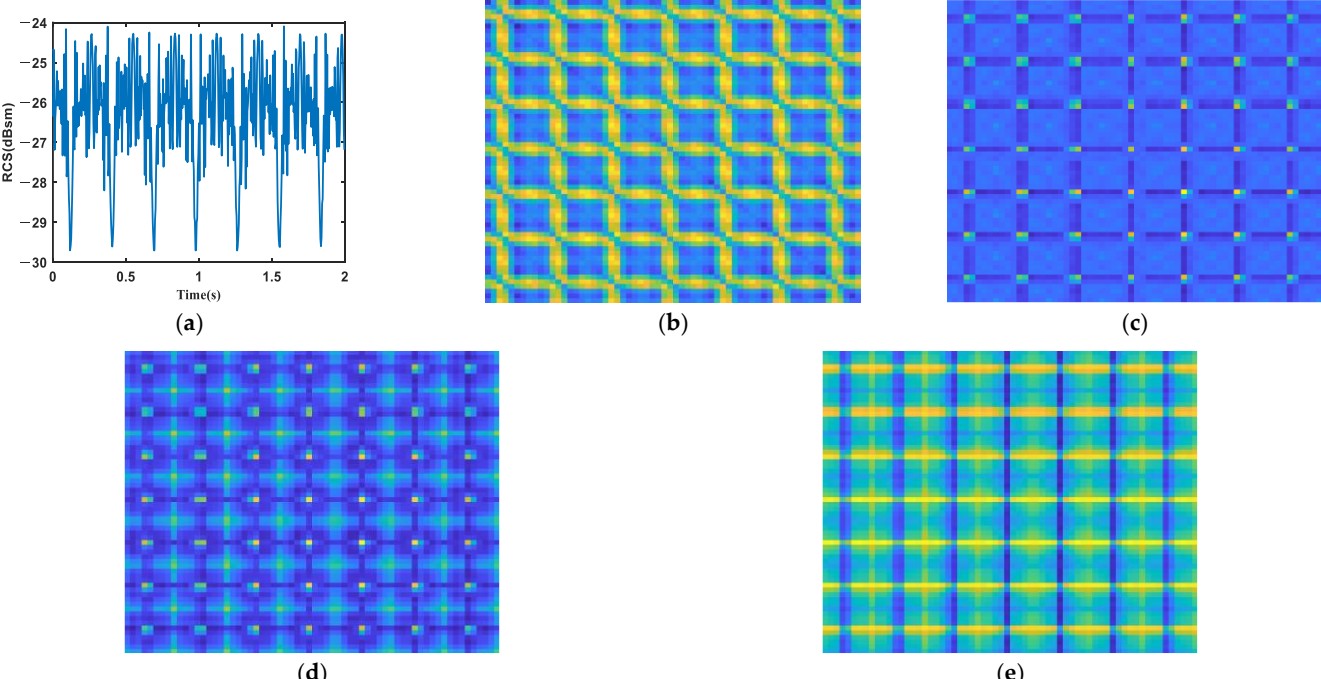

**Figure 4.** (**a**) RCS; (**b**) RP; (**c**) MTF; (**d**) GASF; (**e**) GADF. RCS sequences encoding of the warhead.

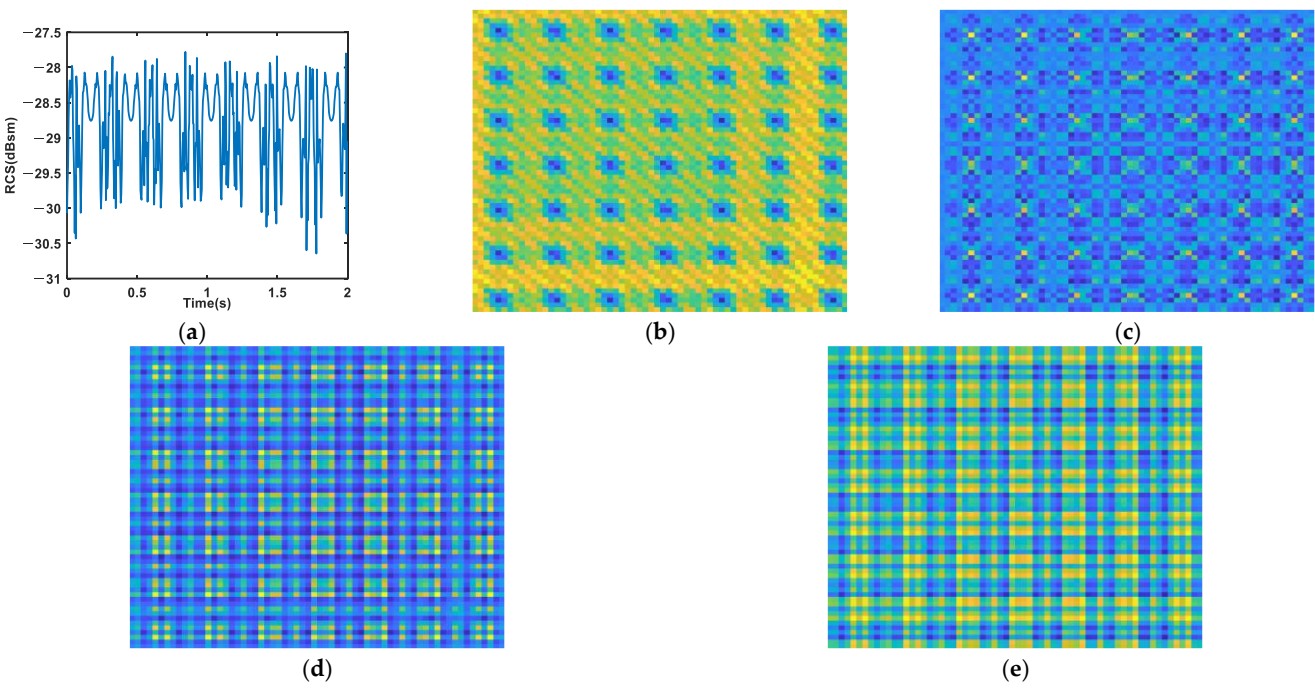

**Figure 5.** (**a**) RCS; (**b**) RP; (**c**) MTF; (**d**) GASF; (**e**) GADF. RCS sequences encoding of the heavy decoy.

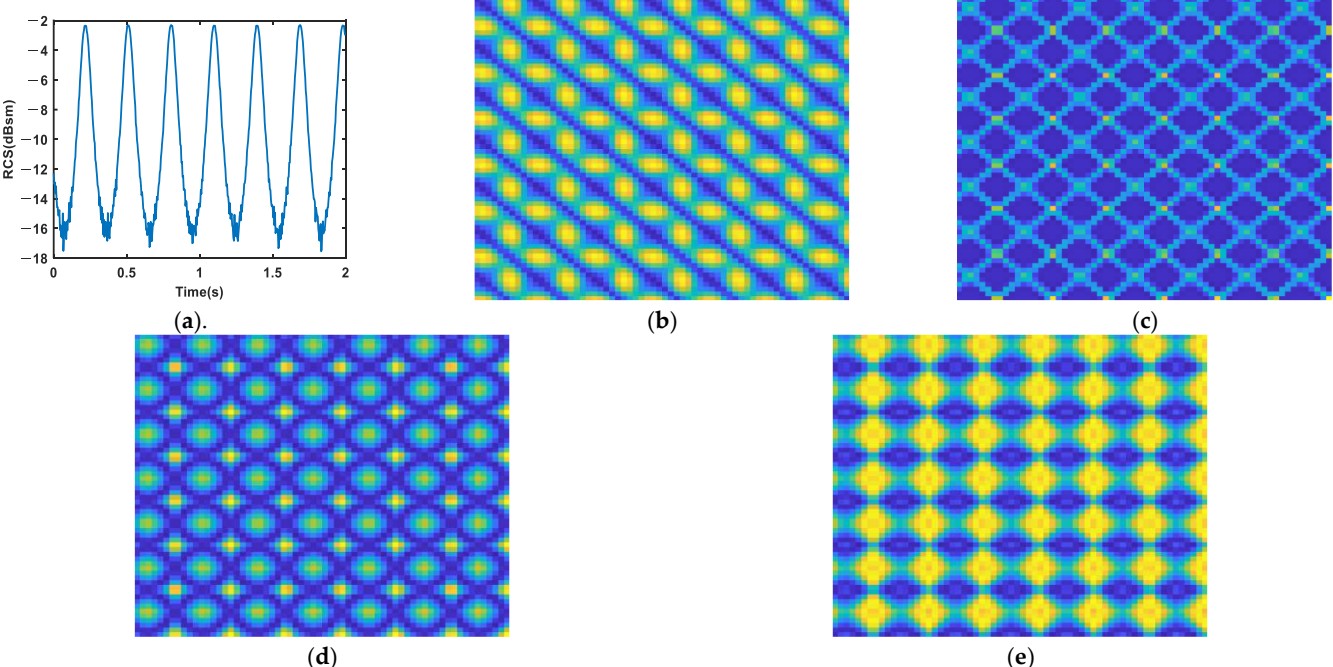

**Figure 6.** (**a**) RCS; (**b**) RP; (**c**) MTF; (**d**) GASF; (**e**) GADF. RCS sequences encoding of the light decoy.

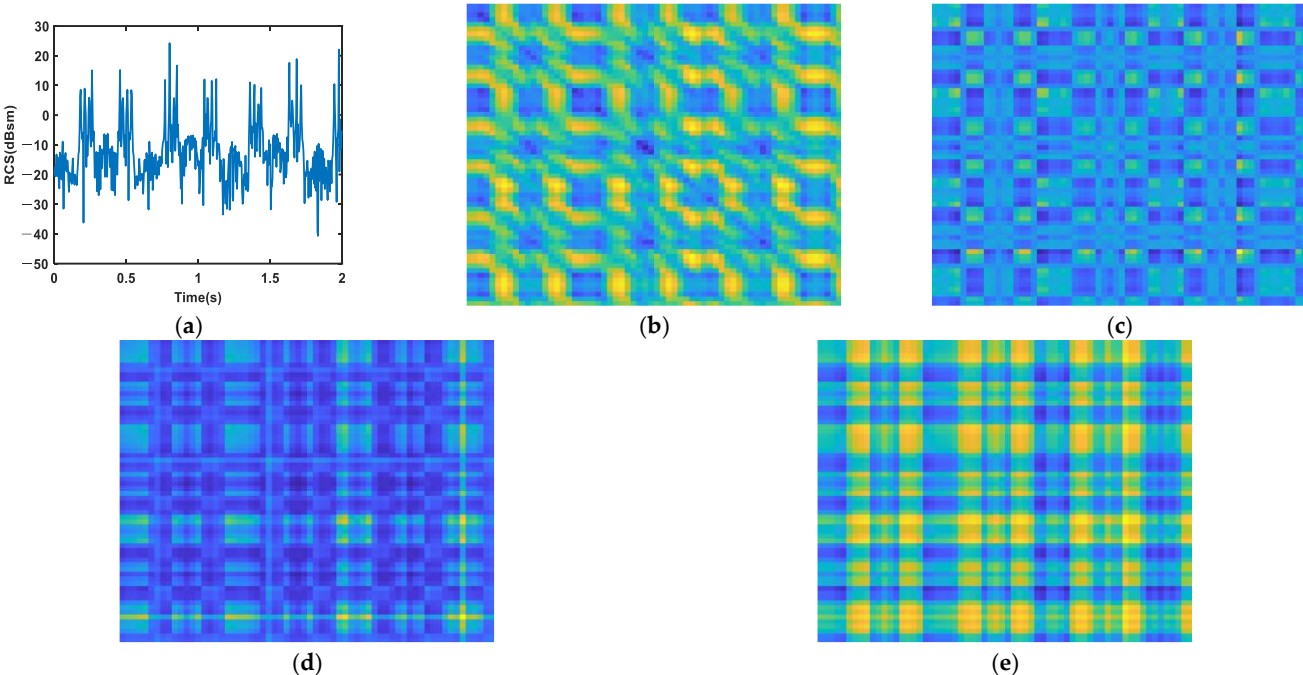

**Figure 7.** (**a**) RCS; (**b**) RP; (**c**) MTF; (**d**) GASF; (**e**) GADF. RCS sequences encoding of the booster.

It needs to be noted that the RP is generated with $\tau = 4$, $m = 5$, and the MTF is generated with $Q = 8$. According to Figures 4–7, we find that different targets will induce different RCS sequences. The RCS of the precession warhead is relatively complex, which means that precession is a complex motion. The RCS of the rolling light decoy is simple, and the period is obvious. The structure of the rolling booster is the most complex among the four targets, and the RCS sequence of the swing weight decoy is relatively complex. As for the encoding methods, the textures on the RP are the sharpest, which means that the chaotic characteristic of the RCS sequence is strong. Textures on the other three images are very irregular, which may induce low classification accuracy.

As for the splitting of the dataset, the random selection method is used to split the dataset to guarantee the generalization ability of the network. We divide the dataset into a training subset (60%), a validation subset (20%) and a test subset (20%). The training subset is adopted to train the network; the validation subset is used to evaluate the training process and select suitable hyperparameters; and the testing subset is adopted to evaluate the performance of the network. After conducting the performance analysis of several training processes with different hyperparameters, the following parameters are selected as the hyperparameters to achieve a relatively high accuracy classification. We use the SGDM method to train the proposed network; the initial learning rate is set to 0.02 and will drop by a factor of 0.5 every 4 epochs; the total epoch is set to 20, and the batch size is set to 32. Moreover, the computer graphics card is NVIDIA GeForce RTX 3070. The hyperparameters of the second and third experiments are the same as those of the first experiment.

Accuracy, Precision, Recall and $F_1$ are the four indicators widely used for evaluating the classification task. The performance analysis of the proposed method is based on these indicators. TP represents how many positive samples predict positively; FP denotes how many negative samples predict positively; TN stands for the number of positive samples, which are predicted to be negative; and FN stands for the number of positive samples whose predictions are negative. These four indicators are solved via Equation (23).

$$
\begin{cases}
\text{Accuracy} = \frac{TP+TN}{TP+TN+FP+FN} \\
\text{Precision} = \frac{TP}{TP+FP} \\
\text{Recall} = \frac{TP}{TP+FN} \\
F_1 \quad = \frac{2 \times \text{Precision} \times \text{Recall}}{\text{Precision}+\text{Recall}}
\end{cases}
\tag{23}
$$

To evaluate the performance of the encoding methods and the proposed network, three group experiments are conducted in this paper. The first experiment is conducted to show the accuracy of the different encoding methods. The result is shown in Tables 3–5.

**Table 3.** The classification evaluation based on MTFs.

| Q | Warhead | Weight Decoy | Light Decoy | Booster | |
|---|---|---|---|---|---|
| | | Recall | | | Accuracy |
| 4 | 0.8854 | 0.7474 | 0.8964 | 0.9125 | 0.8604 |
| 8 | 0.9219 | 0.7604 | 0.9115 | 0.9427 | 0.8841 |
| 16 | 0.9083 | 0.7724 | 0.8828 | 0.9323 | 0.8740 |
| 32 | 0.8906 | 0.7380 | 0.8573 | 0.8875 | 0.8434 |
| 64 | 0.7953 | 0.6510 | 0.8224 | 0.7917 | 0.7651 |

**Table 4.** The classification evaluation based on GAFs.

| | Warhead | Weight Decoy | Light Decoy | Booster | |
|---|---|---|---|---|---|
| | | Recall | | | Accuracy |
| GASF | 0.9323 | 0.7917 | 0.9115 | 0.9740 | 0.9023 |
| GADF | 0.9167 | 0.7708 | 0.9479 | 0.9167 | 0.8880 |

**Table 5.** The classification evaluation based on RPs.

| ($\tau$,$m$) | Warhead | Weight Decoy | Light Decoy | Booster | |
|---|---|---|---|---|---|
| | | Recall | | | Accuracy |
| (1, 1) | 0.9573 | 0.8891 | 0.9599 | 0.9938 | 0.9500 |
| (1, 3) | 0.9526 | 0.8844 | 0.9620 | 0.9923 | 0.9478 |
| (1, 5) | 0.9635 | 0.8734 | 0.9609 | 0.9932 | 0.9478 |
| (4, 3) | 0.9547 | 0.8734 | 0.9740 | 0.9953 | 0.9493 |
| (7, 3) | 0.9620 | 0.8990 | 0.9484 | 0.9953 | 0.9512 |
| (4, 5) | 0.9635 | 0.9010 | 0.9531 | 0.9792 | 0.9492 |
| (4, 7) | 0.9672 | 0.8776 | 0.9599 | 0.9958 | 0.9501 |

Table 3 presents the performance of the proposed network based on MTF. Different values of $Q$ mean different numbers of the quantile bins. With the increase in $Q$, the accuracy goes up first but then goes down. This tendency indicates that either too large or too small $Q$ is unfavorable for feature expression, and thus, the poor accuracy occurs. When $Q$ is set to 8, the accuracy is the best, and the recall for the three targets is also the best. Setting $Q$ to 8 is the most reasonable choice for MTF-based classification methods.

According to Table 4, although there are differences in the discrimination ability for different targets, the overall classification ability of GASF and GADF is similar. We think that the reason for this phenomenon is that the generation ways of GASF and GADF are similar.

Seven groups with different $\tau$ and $m$ are conducted to generate different RPs. Overall, the difference will induce a different accuracy for the proposed network, but the difference is not obvious. The difference between the best accuracy and the worst accuracy is only 0.0034. However, compared with MTF and GAF, the RP-based network performs much better than the other two encoding methods. The difference between the RP-based network and the other two networks is approximately 0.06, and such difference is significant. Such a result is consistent with our previous statement, as RP intuitively displays more details of RCS sequences than other methods.

To make the classification results more accurate and clear, the confusion matrices of different encoding methods are presented in Figure 8. It needs to be noted that the confusion matrices give a synthesis of the classification results obtained from 10 trials.

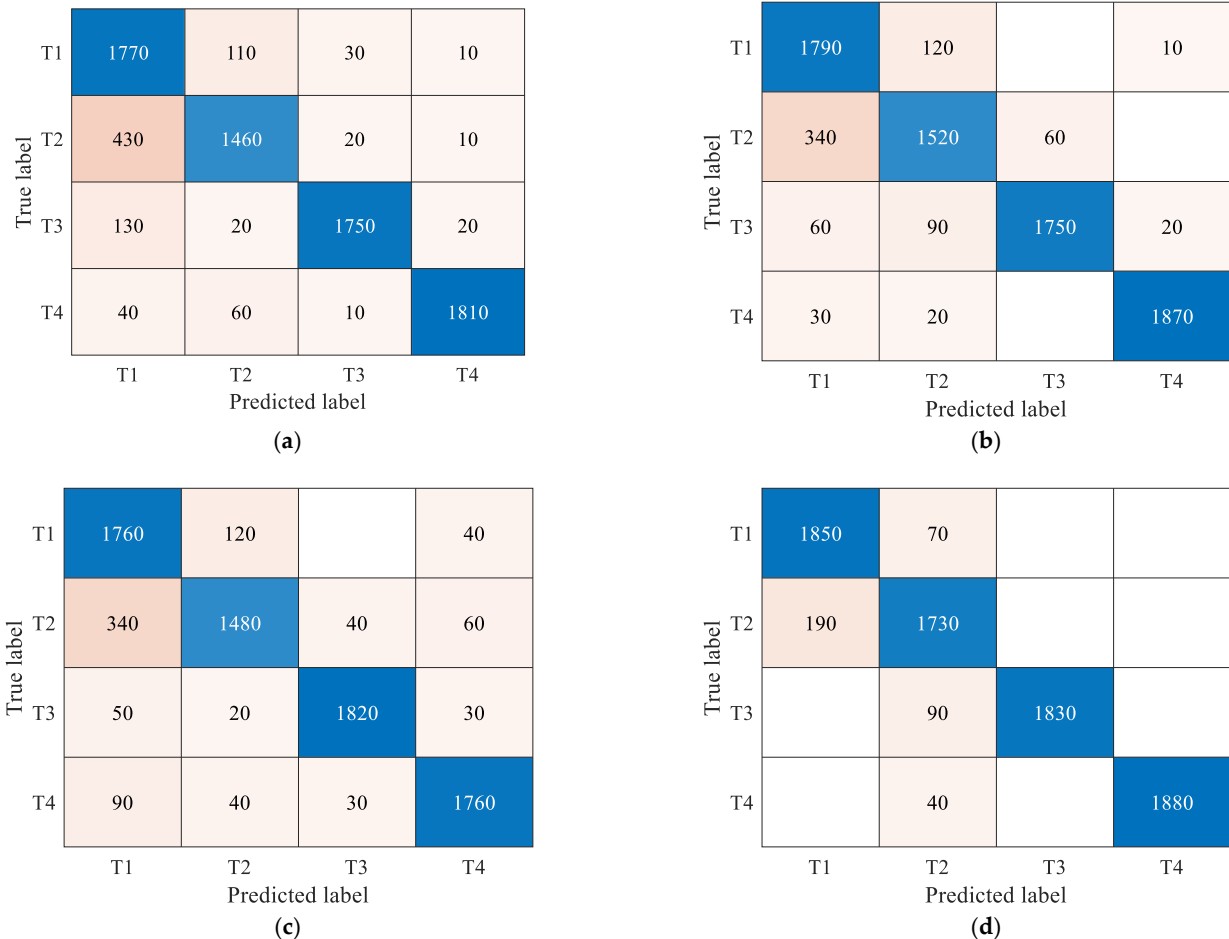

**Figure 8.** (**a**) MTF-8; (**b**) GASF; (**c**) GADF; (**d**) RP (4, 5). Confusion matrices of different encoding methods.

In Figure 8, T1, T2, T2 and T4 represent the warhead, the heavy decoy, the light decoy and the booster, respectively. According to the accuracy of these three encoding methods, we draw the conclusion that RP is the optimal encoding method, GAF is the second, and MTF is the worst. For these four targets, the booster is the easiest to identify; the classification difficulties of the light decoy and the warhead are close; and the weight decoy is the most difficult to identify.

The second experiment is conducted to evaluate how the proportion of the training set affects the classification accuracy. In general, we believe that the number of training sets will have an impact on the performance of the network. As a result, we conduct the experiment to analyze how strong the impact is. Taking the RP as the input, different proportions of the training set to the total samples are employed to train the network, and the accuracy of the test set is shown in Figure 9.

Since this experiment aims to analyze the effects of the number of training sets, all the experiments should be performed with the same hyperparameters. As a result, the parameters are kept the same as those in the first experiment. Moreover, we only split the datasets into the training subset and the testing subset, without taking the validation subset into consideration. With the proportion value changing from 0.1 to 0.8, the accuracy of the proposed network is constantly increasing. This is a normal trend because the larger the number of training sets, the richer the features the network can learn. However, when the proportion changes from 0.8 to 0.9, there is a small drop emerging in the accuracy. This phenomenon is very strange and incredible. After consulting several literature works, we found a reasonable explanation for this phenomenon. More training sets mean fewer test sets are available, which makes it difficult for the network to match the test set with the prediction labels. Within certain limits, the impact of a small test set is stronger than the

impact of a large training set. This conclusion is very important, and it will provide an important reference for our future work.

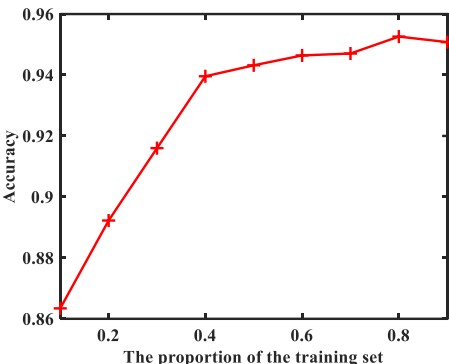

**Figure 9.** The performance with different proportions of the training set.

To evaluate the robustness of the proposed method, experiments are conducted with different SNRs. Furthermore, three CNNs are employed as comparisons to verify the superiority of the proposed method. It should be noted that the proposed method, Resnet50 and Alexnet are based on the RP ($\tau = 4$, $m = 5$), while the 1D CNN is based on the RCS sequences. Based on the optimal result of the second experiment, the proportion of the training set is set to 0.8, and the proportion of the testing set is set to 0.2. By using the random selection method, the dataset for the third experiment is generated. Taking the accuracy and F1 score as the criteria, the results are shown in Figure 10 and Table 6.

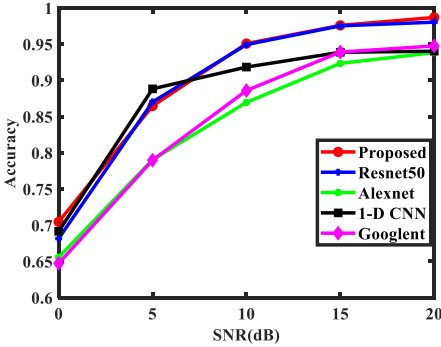

**Figure 10.** The classification performance for different methods.

**Table 6.** The performance evaluation based on F1 score.

|  | **20 dB** | **15 dB** | **10 dB** | **5 dB** | **0 dB** |
|---|---|---|---|---|---|
| Resnet50 | 0.9802 | 0.9752 | 0.9493 | 0.8697 | 0.6822 |
| Googlenet | 0.9470 | 0.9387 | 0.8847 | 0.7838 | 0.6306 |
| Alexnet | 0.9373 | 0.9226 | 0.8672 | 0.7845 | 0.6387 |
| 1D CNN | 0.9390 | 0.9370 | 0.9153 | 0.8815 | 0.6938 |
| Proposed | 0.9868 | 0.9758 | 0.9507 | 0.8639 | 0.7013 |

According to the accuracy and F1 score, the algorithm in this paper has the best performance among the four algorithms in most cases. The advantages of the algorithm in this paper are mainly reflected in two aspects. (1) Compared with AlexNet, Resnet50 and Googlenet, the datasets of the proposed algorithm are the same as those in the three networks, but the network structure is different. A comparison of these three algorithms shows the effectiveness of the proposed network. (2) Compared with 1D CNN, the superiority of the proposed method in this paper is obvious, which indicates that compared with the original RCS sequence, the encoding of the RCS sequence can also improve the recognition

rate of the image. However, there is one factor, which is common to all algorithms. When the SNR is low, the performance of the RCS-based method is relatively poor, which proves that RCS is sensitive to the noise.

In addition to the accuracy, time consumption is also an important indicator for performance analysis. We calculate the prediction time for one sample of the five methods, and the result is presented in Table 7.

**Table 7.** The performance evaluation based on prediction time for one sample.

|  | Resnet50 | Googlenet | Alexnet | 1D CNN | Proposed |
|---|---|---|---|---|---|
| Time(us) | 17.38 | 12.66 | 5.3701 | 4.3546 | 33.79 |

As shown in Table 7, the prediction time of the proposed method is the highest among the five methods. Obviously, the high accuracy comes at the cost of the network's prediction time for the proposed method. As a result, one valuable aim is to reduce the prediction time of the proposed method.

## 6. Conclusions

In this paper, we develop a framework for BTs classification. The framework first converts the RCS sequences into images with three methods and then inputs those images to the proposed multi-scale CNN. The experimental results show that the RP-based method has the best accuracy among these three encoding methods. Moreover, the proposed network outperforms other existing networks with better accuracy and robustness. However, several parameters are empirically selected, including the parameters involved in the encoding methods and hyperparameters of the network. If those parameters are selected more precisely and scientifically, the performance of the proposed method may benefit from a significant improvement. In our future work, we will study the method to adaptively select those parameters to make the proposed method more effective. However, the performance of the presented network is based on a large number of labeled samples, which is rarely applicable in the real-world radar target classification tasks [42]. As such, developing a few-shot learning method for radar target classification will be a meaningful and promising technique [43,44]. In the future, we will pay more attention to the few-shot learning methods to make the proposed methods more available.

**Author Contributions:** Writing—original draft, X.X.; Writing—review & editing, X.X., C.F. and L.H. All authors have read and agreed to the published version of the manuscript.

**Funding:** This work was supported in part by the Natural Science Basic Research Program of Shaanxi (No. 2021-JQ-361).

**Data Availability Statement:** Not applicable.

**Conflicts of Interest:** The authors declare no conflict of interest.

## Appendix A. Depthwise Separable Convolution (DS-Conv)

In 2017, F. Chollet of Google Labs proposed a new CNN named Xception [45]. The most important highlight of Xception is DS-Conv, which reduces the complexity of the network without a loss of accuracy.

DS-Conv replaces regular convolution with depthwise convolution followed by pointwise convolution (standard convolution with $1 \times 1$ kernel) [46]. First, every channel of the input is convoluted through depthwise convolution, so that spatial correlations are achieved. Then, the features of each channel are combined through a pointwise convolution, and channel correlations are achieved. Set the kernel size $K \times K$ for DS-Conv, and the diagram is shown in Figure A1.

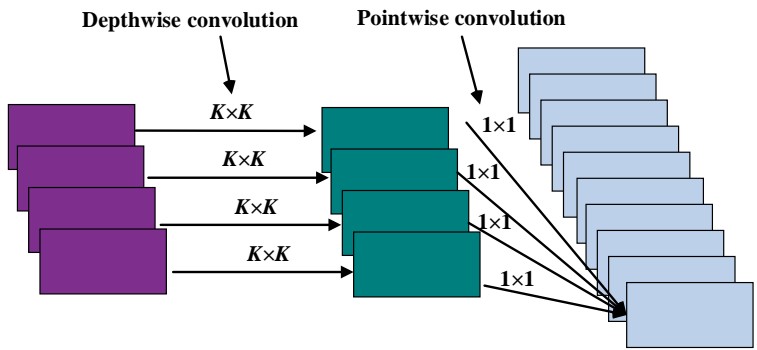

**Figure A1.** The diagram of DS-Conv.

Mathematically, depthwise convolution is formulated as

$$G(x_d, y_d, j_d) = \sum_{u=1}^{k} \sum_{v=1}^{k} K(u, v, j_d) \times F_{in}\left(x_d + u - \frac{k+1}{2}, y_d + v - \frac{k+1}{2}, j_d\right) \tag{A1}$$

where $K$ represents the convolution operation, $k$ denotes the kernel size, $F_{in}$ represents the input ($h \times w \times c_{in}$). $x_d$ and $y_d$ is the position of every pixel, $u$ and $v$ is the step order of the convolution kernel, $j_d$ is the order of the channel, and $G$ is the output feature map. The parameter quantity involved in depthwise convolution is $K \times K \times c_{in}$.

Then, pointwise convolution is calculated as

$$F_{out}(x_d, y_d, l) = \sum_{j=1}^{c_{in}} G(x_d, y_d, j_d) \times P(j_d, l) \tag{A2}$$

where $P(j_d, l)$ is the $1 \times 1$ convolution, $l \in [1, c_{out}]$. The number of parameters involved in this process is $1 \times 1 \times c_{in} \times c_{out}$. For regular convolution, the parameter quantity is $K \times K \times c_{in} \times c_{out}$ [46], which is bigger than that of DS-Conv' ($c_{in} \times c_{out} + K \times K \times c_{in}$).

To reduce the parameter quantity, this paper uses separable convolution many times in the proposed network.

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
