# Peer review of "Classification of Radar Targets with Micro-Motion Based on RCS Sequences Encoding and Convolutional Neural Network"

_remotesensing, doi:10.3390/rs14225863_

Round 1

Reviewer 1 Report

In this paper, a multi-scale CNN-based framework is proposed for classification radar targets with Micro-motion. By exploring three types of features with Gramian Angular Field, Markov Transition Field and Recurrence Plot, the proposed method can achieve an excellent result. Specifically, the reviewer has the following comments to this work:

1.       More experiment results should be added, such as confusion matrix, which is able to evaluate the network’s ability of detection.

2.       According to the paper, the proposed network seems to have many trainable parameters, which leads to time consuming. It is recommended to add a comparison of time consumption between different methods

3.       The layout of this article is confusing, with the same fig and table scattered across different pages.

4.       In the introduction section, some important references on recognition of targets with micromotion are omitted. It is recommended to add these works to the reference, such as:

(1)    https://www.doi.org/10.1109/TAES.2022.3145303

(2)    https://www.doi.org/10.1109/TAES.2010.5595607

5.       The CNN-based methods usually need a lot of training samples. The few-shot learning method can effectively reduce the demand for the number of training samples and has achieved great success in classification problems. It is recommended to add these works to the reference, such as:

(1)    https://www.doi.org/10.1016/j.displa.2022.102162

(2)    https://www.doi.org/10.1109/TVT.2022.3196103

(3)    https://www.doi.org/10.1109/TCSVT.2021.3088545

Author Response

The Responses can be found in the attachment.

Reviewer 2 Report

The authors have have used Radar Cross Section to classify different targets using Convolutional Neural Network

  • . I have a few questions to ask.
  • The accuracy difference between ResNet50 and the proposed is not too significant. Add comments about the computational overhead. Is the proposed system more computationally effective than ResNet50?
  • how about comparing with 2D CNN using radar echoes?
  • Below paper used a variant of CNN architecture. Authors can mention this study in the introduction to show similar radar target recognition works . Noori, F. M., Riegler, M., Uddin, M. Z., & Torresen, J. (2020). Human activity recognition from multiple sensors data using multi-fusion representations and CNNs. ACM Transactions on Multimedia Computing, Communications, and Applications (TOMM)16(2), 1-19.

Author Response

(The authors gave the same response as above.)

Reviewer 3 Report

The paper is well written and exposes a novel method to classify targets based on the radar cross-section. 

To improve the quality of the paper I suggest improving the explanation of how the dataset is built and composed. 

The main concern about the work is the methodology. Below you can find other my comments. The paper is well written and exposes a novel method to classify targets based on the radar cross-section. To improve the quality of the paper I suggest improving the explanation of how the dataset is built and composed. How the authors build the signals used for the classification is not clear. Some choices must be motivated such as how they choose the dimension of the dataset, and how they split them for training and test. Equation 1 must be referenced. The quality of Fig. 4 is too low. Please consider a better resolution.

Author Response

(The authors gave the same response as above.)

Reviewer 4 Report

The paper is well-written. However, I have concerns about the contributions. I understand that the authors are using existing methods to convert the RCS sequences to images and then use existing networks to classify the radar targets. All these techniques already exist, and their application is not a sufficient contribution to be acceptable as a journal paper. If there are some novel contributions of the authors, they should be pointed out. In addition, a lot of description is given for these already existing techniques, which is not necessary and can be put in an Appendix. 

I have a few other comments: 

1) What is the reason for selecting the configuration shown in Table 1? How did the authors obtain it? Why not select another configuration? What is the reason for selecting Resnet? Why not use Alexnet or another configuration?

2) How were the hyperparameters of the neural networks chosen?

3) Please correct the spellings of "Conclusioin".

Author Response

(The authors gave the same response as above.)

Round 2

Reviewer 1 Report

The author has revised the manuscript according to the comments of the reviewers. I have no other questions.

Author Response

Dear reviewer 1

Thank you very much for your contribution to this paper.

Sincerely,

Xuguang Xu, Cunqian Feng

Reviewer 2 Report

My comments have been answered

Author Response

Dear reviewer 2

Thank you very much for your contribution to this paper.

Sincerely,

Xuguang Xu, Cunqian Feng

Reviewer 4 Report

Please compare the results with non-encoding based methods, in order to show that the encoding really improves the performance. Please also give a thorough editing to the paper to correct for language mistakes.

Author Response

Dear reviewer 4

Thank you very much for your contribution to this paper.  We have performed a thorough English proofreading on this paper. As for the non-encoding based methods, the experiment of 1-D CNN is based on the raw RCS sequences(without encoding). And the poor performance of 1-D CNN proves the advantages  of the encoding methods. 

Sincerely,

Xuguang Xu, Cunqian Feng